# The In Vitro Protective Role of Bovine Lactoferrin on Intestinal Epithelial Barrier

**DOI:** 10.3390/molecules24010148

**Published:** 2019-01-02

**Authors:** Xiao Zhao, Xiao-Xi Xu, Yang Liu, En-Ze Xi, Jing-Jing An, Dina Tabys, Ning Liu

**Affiliations:** 1Key Laboratory of Dairy Science, Ministry of Education, Northeast Agricultural University, Harbin 150030, China; zhaoxiao1816@neau.edu.cn (X.Z.); tabysdina@gmail.com (D.T.); liuning@neau.edu.cn (N.L.); 2College of Food Science, Northeast Agricultural University, Harbin 150030, China; Xienze1993@163.com (E.-Z.X.); anjingloveme@163.com (J.-J.A.); 3China Feihe Co. Ltd., Beijing 100015, China; liuyang1@feihe.com

**Keywords:** bovine lactoferrin, tight junction protein, intestinal epithelial barrier, human intestinal epithelial crypt cells, permeability

## Abstract

The intestinal epithelial barrier plays a key protective role in the gut lumen. Bovine lactoferrin (bLF) has been reported to improve the intestinal epithelial barrier function, but its impact on tight junction (TJ) proteins has been rarely described. Human intestinal epithelial crypt cells (HIECs) were more similar to those in the human small intestine, compared with the well-established Caco-2 cells. Accordingly, both HIECs and Caco-2 cells were investigated in this study to determine the effects of bioactive protein bLF on their growth promotion and intestinal barrier function. The results showed that bLF promoted cell growth and arrested cell-cycle progression at the G2/M-phase. Moreover, bLF decreased paracellular permeability and increased alkaline phosphatase activity and transepithelial electrical resistance, strengthening barrier function. Immunofluorescence, western blot and quantitative real-time polymerase chain reaction revealed that bLF significantly increased the expression of three tight junction proteins—claudin-1, occludin, and ZO-1—at both the mRNA and protein levels, and consequently strengthened the barrier function of the two cell models. bLF in general showed higher activity in Caco-2 cells, however, HIECs also exhibited desired responses to barrier function. Therefore, bLF may be incorporated into functional foods for treatment of inflammatory bowel diseases which are caused by loss of barrier integrity.

## 1. Introduction

Bioactive protein molecules are widely found in animals, plants, and microorganisms [1]. These proteins and their hydrolysates have antimicrobial, mineral-binding, antihypertensive, antioxidant, anticancer, and other bioactivities, providing health benefits and promoting well-being [2,3]. Milk is nutritionally balanced and is a good source of bioactive proteins [4]. Bovine lactoferrin (bLF) is a glycoprotein found in milk and is present in most exocrine secretions including tears, saliva, intestinal mucus, genital secretions, and the specific granules of neutrophils. Studies have demonstrated that bLF has antibacterial, immune-modulating, and anti-inflammatory properties [5,6], and notably, protective effects on intestinal barrier integrity [7]. Addition of human LF to lipopolysaccharide-injured Caco-2 cells revealed that this LF improves impaired barrier function based on transepithelial electrical resistance (TEER) and the permeation of fluorescein isothiocyanate—labeled dextran 4000 [8]. A recent study used indicated that bLF improved intestinal barrier integrity in rats subjected to bowel resection [9], possibly owing to up-regulated claudin-4 gene expression [10]. However, it remains unclear whether bLF can strengthen intestinal barrier function in other epithelial cells such as human intestinal epithelial crypt cells (HIECs).

Comprehensive intestinal barrier function is critical to body health. A complete multilevel barrier system is essential for maintaining the dynamic balance between the organism and intestinal tract. The intestinal mucosa forms a physical and metabolic barrier against the diffusion of pathogens, toxins, and allergens from the lumen into the circulatory system [11]. Impaired barrier function of the intestinal mucosa increases host susceptibility to luminal antigens and pathogens, followed by a chronic response of the intestinal immune system [12]. Accordingly, intestinal barrier dysfunction may destroy immune homeostasis and induce an inflammatory response [13]. The formation of tight junctions (TJs) in epithelial cells plays a pivotal role in the intestinal barrier [14]. TJs mediated by proteins such as claudins, occludin, and zonula occludens (ZO) are necessary for epithelial barrier maintenance [15,16], disruption of the intestinal epithelial barrier can increase intestinal permeability [17].

TEER is a classic indicator of the strength of TJs and reflects the ionic conductance of the paracellular pathway in cell monolayers, whereas the flux of nonelectrolyte compounds indicates the paracellular water flow and pore size of TJs [18]. Moreover, expression changes in TJ proteins are crucial and typically assessed to evaluate the barrier function of epithelial monolayers. Behaving as multiple protein complexes, TJs form a selectively permeable seal between adjacent epithelial cells [19]. Occludin was the first integral membrane TJ protein to be discovered in 1993 [20]. Although the roles of occludin in TJs require further analysis, occludin appears to be closely associated with other TJ proteins, and thus plays essential roles in TJ function [21]. The claudin family contains at least 20 members that are considered the principle barrier-forming proteins [22,23]. ZO proteins, members of the junctional adhesion molecule family, maintain the TJ structure and modulate barrier integrity by interacting with the actin cytoskeleton [24]. Increasing evidence has demonstrated that increased intestinal permeability plays a pathogenic role in intestinal diseases, such as inflammatory bowel disease and celiac disease, and functional bowel disorders, such as irritable bowel syndrome [25,26]. Interestingly, many TJ molecules in endothelial cells of the blood-brain barrier are identical to those in intestinal tissues, such as occludin, claudins, and ZO-1 [16]. Therefore, protecting TJ integrity is important for intestinal epithelial barrier function.

Several cell lines derived from human or animal intestine have been used to study the characteristics and functions of the human intestine. Caco-2 cells are the “gold standard” of these models [27], and have been widely used to investigate cell differentiation and its regulatory mechanisms, the influence of growth factors, absorption and metabolism of drugs [27,28]. HIECs were derived from the small intestine of a healthy neonate, and are considered a useful cell model for analyzing the regulation of cell growth and differentiation and determining the functional basis of cell matrix interactions [29]. Recent studies showed that HIECs can be used to precisely predict the absorption of compounds and microRNA responses in humans [30,31]. The components specific to desmosomes and TJs of HIECs [28] indicate that these cells are promising tools for the study of intestinal barrier function. However, to the best of our knowledge, no study has investigated the response of HIECs to bLF. Furthermore, HIECs may be a better cell model than Caco-2 cells that more accurately represents human intestinal barrier function, but this remains unverified.

Thus, this study was conducted to assess the response of Caco-2 and HIECs to bLF stimulation, using cell viability, cell-cycle progression, TEER, monolayer permeability, and TJ protein expression as evaluation indices. We hypothesized that bLF would play a beneficial role on the intestinal epithelial barrier and HIEC would be an ideal cell mode to be used in human intestinal epithelial barrier function study.

## 2. Results

### 2.1. Growth Promotion Effect of bLF on the Two Cell Lines

Five doses of bLF were used to determine its effect on cell growth at three cell culture times (Figure 1). The results indicated that bLF promoted the growth of the two cell line at each time point, resulting in increased cell viability (i.e., higher than 100%). Overall, bLF led to 2.6–40.3% and 3.2–33.7% increases in viability Caco-2 cells and HIECs, respectively. Thus, HIECs were less sensitive than Caco-2 cells to bLF. bLF doses of 50, 100, and 200 μg/mL resulted in higher viability values; however, a bLF dose of 200 μg/mL yielded viability increases of only 3.8% and 4.0% at 48 h compared with a bLF dose of 100 μg/mL in Caco-2 cells and HIECs, respectively. Therefore, bLF was used at 50 and 100 μg/mL in subsequent experiments. A culture time of 2 days led to higher growth promotion, and was thus used in later cell experiments.

### 2.2. Cell-Cycle Distribution in the Two Cell Lines Treated with bLF

Epithelial monolayers were pretreated with bLF at different doses, and the cell-cycle distribution was determined by flow cytometry (Figure 2).

In Caco-2 control cells, the respective portions of cells in the G0/G1-, S-, and G2/M-phases were 51.9%, 30.6%, and 17.5%. After 48 h of exposure to bLF at 50 μg/mL, the percentage of Caco-2 cells in the G0/G1 phase was significantly reduced (45.4%), along with an increased cell percentage in the S- and G2/M-phases (32.2% and 22.4%). A bLF dose of 100 μg/mL considerably increased the proportion of cells in the S- and G2/M phases (35.3% and 23.9%) and reduced the cell percentage in G0/G1 phase (40.8%). In HIECs, bLF at 50−100 μg/mL showed similar effects, the cell percentage in the S- and G2/M-phases was increased to 23.0–27.1% and 8.6−9.6%, respectively, whereas that in the G0/G1-phase was decreased to 66.4−63.3%. Overall, bLF arrested the cell cycle at the G2/M-phase, which promoted cell growth (or higher cell viability).

### 2.3. Cell Differentiation of the Two Cell Lines Treated with bLF

After a total culture time of 21 days and evaluation at three time points, ALP activity was measured as an evaluation index. bLF at two different doses induced the differentiation of Caco-2 cells and HIECs (Figure 3). The ALP activity values in Caco-2 cells were increased to 138−200 mU/mg after culture for 14 days but decreased to 120−165 mU/mg after culture for 21 days. ALP activity values in HIECs were 114−150 mU/mg but decreased to 108−132 mU/mg at the same time points. However, a culture time of 7 days did not enable assessment of differences between the two cell monolayers after treatment with bLF, as the ALP values were not significantly different (*p* > 0.05). Notably, 100 μg/mL bLF treatment resulted in higher ALP activity than 50 μg/mL bLF in the two cell lines. In addition, Caco-2 cells showed higher differentiation potential than HIECs, as Caco-2 cells showed higher ALP activity at 14 and 21 days.

### 2.4. Effects of bLF on Epithelial Monolayer Resistance and Permeability of Two Cell Lines

Compared with the untreated cells, both Caco-2 and HIEC monolayers treated with bLF showed significantly increased TEER values (*p* < 0.05, Figure 4A), indicating an improvement of TJs.

Treatment with 50 and 100 μg/mL bLF increased TEER values by 17% and 41%, respectively, in Caco-2 monolayers, and by 31% and 65%, respectively, in HIEC monolayers. Interestingly, the TEER values of Caco-2 monolayers were much higher than those of HIEC monolayers.

The results in Figure 4B,C showed the transport of sodium fluorescein across Caco-2 cell and HIEC monolayers in the presence and absence of bLF. The apparent permeability coefficient (*P*_app_) value of the control Caco-2 cell monolayers did not show an obvious increase in the first 30 min, but showed a 200% increase at 60 min. A longer incubation time of 120 min further increased the *P*_app_ value (Figure 4B). Caco-2 cell monolayers exposed to bLF showed similar changes in the *P*_app_ value, but always had a lower *P*_app_ value (especially when 100 μg/mL bLF was used) (Figure 4B). Similar results were obtained when HIEC monolayers were assessed (Figure 4C). All results suggested that bLF decreased the transport of sodium fluorescein dose-dependently in the two cell monolayers, thereby improving the physical epithelial barrier.

### 2.5. Effect of bLF on Expression of TJ Proteins

The qRT-PCR results showed that 50 and 100 μg/mL bLF up-regulate the mRNA expression of *CLDN-1*, *OCLN*, and *TJP-1* in the treated cells relative, to the untreated control cells (Figure 5A).

In Caco-2 cells, 50 μg/mL bLF enhanced *CLDN-1*, *OCLN*, and *TJP-1* expression levels by 1.32-, 1.26-, and 1.30-fold, while 100 μg/mL bLF increased these expression levels by 2.45-, 1.65-, and 1.82-fold. In HIEC, bLF at 50 and 100 μg/mL increased *CLDN-1*, *OCLN*, and *TJP-1* expression levels by 2.14–2.57-, 1.03–1.22-, and 1.04–1.69-fold, respectively. Considering the critical roles of the three TJ genes, the present results implied that the barrier function in the two cell monolayers was improved.

To further verify the beneficial effect of bLF on intestinal barrier function, immunofluorescence and western-blot (WB) were performed to assay the expression of TJ proteins in the two cell models. The cell monolayers precultured with bLF showed increased expression of claudin-1, occludin, and ZO-1. bLF at 50 μg/mL increased the expression levels of the three TJ proteins by 1.48–1.87-fold in the Caco-2 cells and, 1.33–1.60-fold in the HIECs (Figure 5B). bLF at 100 μg/mL induced higher expression levels of these TJ proteins (1.65–2.45-fold for Caco-2 cells, and 1.22–2.57-fold for the HIECs) (Figure 5C). Immunofluorescence assay results indicated that all three TJ proteins were distributed along the cell membrane, and these distribution features were entirely observed as honeycomb linear fluorescence (Figure 6). In this assay, a dose dependent response of expression of the three TJ proteins to bLF was also observed, based on the increased red fluorescence intensity. Overall, the results directly demonstrated that bLF has the ability to increase TJ protein production and thereby strengthen barrier function.

## 3. Discussion

Intestinal barrier integrity is a key feature in the health of humans, particularly newborns, because the immature gastrointestinal system and neonatal immune system are still developing [32]. Thus, barrier integrity is of primary importance to prevent the passage of noxious agents from the intestinal lumen into the mucosa and blood circulation [33]. If this barrier integrity is disturbed, allergic diseases such as food allergy, asthma, and inflammatory bowel diseases can be triggered [24,34,35]. Patients with Crohn’s disease show increased intestinal permeability, inducing defective barrier integrity [13]. Patients suffering from birch pollinosis and other food allergies are also diagnosed with increased intestinal permeability [36,37,38]. The intestinal barrier is thought to greatly depend on the formation of specialized intercellular TJs. In previous studies, several compounds were shown to increase barrier integrity in Caco-2 and HT29 cells [30,39]. For, example, quercetin is widely found in fruits and has been demonstrated to enhance barrier function by improving the expression of claudin-4 in Caco-2 cells [40]. These results support those of the present study that up-regulate the expression of TJ protein can improve the intestinal barrier function.

Dietary components have the most pronounced effect on the intestinal environment because they regulate the intestinal barrier and exert other biofunctions. Numerous studies focused on the role of food components in intestinal permeability to identify the potential correlations between diet, intestinal permeability, obesity, and inflammatory bowel disease [41,42]. Supplementation of the diet of pigs with 2 g/kg basal diet or more of zinc oxide enhanced the expression of occludin and ZO-1 and increased the TEER value of the small intestine [43]. In the rat model of hemorrhagic shock followed by resuscitation, ω-3 polyunsaturated fatty acids were found to decrease TJ disruption in the mucosa and increase the expression of occludin in the jejunum mucosa [44]. Similar to the results of these studies, treatment of IEC-6 cells with grape-seed procyanidins can up-regulate the expression of occludin and ZO-1, thereby reducing cell permeability [45]. Another example is herb-partitioned moxibustion, which can promote occludin, claudin-1, and ZO-1 expression, and recover the increased epithelial permeability in Crohn’s disease model rats [46]. Thus, it is reasonable that bLF had the ability to increase claudin-1, occludin, and ZO-1 TJ protein expression, effectively strengthening barrier function of the two cell models used in this study.

bLF is an important bioactive protein in the mammal secretion, which is relatively resistant to proteolytic enzymes [47], is thus not totally degraded in the luminal content of stomach and small intestine allowing binding to specific receptors located in the brush border membranes vesicles origined from fetal intestine [48]. LFRs specifically mediate the uptake of bLF into enterocytes and crypt cells. bLF transported from the intestinal lumen to the bloodstream and function not only at the luminal intestinal level but also systemically [49]. Our results showed bLF increased proliferation of both Caco-2 cells and HIECs, which was associated with the arrest of the cell cycle at G2/M and inhibited apoptosis. Similar results were found that LF up-regulate Caco-2 growth through decreased cell spontaneous apoptosis [50].

bLF might play beneficial role on the intestinal epithelium barrier through NK-κB signal pathway. The NF-κB signaling pathway plays an important role in the inflammatory and immune system regulation [51]. bLF could regulate the inflammatory and immune system. TNF-α is one of the important activator of NF-κB signaling pathway. bLF significantly decreased the mRNA expression of TNF-α in a dose-dependent manner in our study, but which needs to be investigated in the further experiment, so data were not show in this paper.

Caco-2 cells are well-established epithelial cells [52] and are widely used as a cell model to investigate the barrier function of the intestinal epithelium. After culture for 21 days, Caco-2 monolayers acquired full morphological polarity with junctional complexes and exhibited similar features to those found in the human fetal intestinal epithelium [53]. HIECs have been recently used by several researchers to assess cell proliferation, cell apoptosis [28], and particularly intestinal permeability [30,54]. However, HIECs are not widely used as a cell model in studies of intestinal TJs. Takenaka et al. suggested that HIEC monolayers are similar to those in the human small intestine [30]. For example, the TEER value of HIEC monolayers (98.9 Ω × cm^2^) was close to that in the human small intestine (40 Ω × cm^2^) [30]. In contrast, the TEER value of Caco-2 monolayers was reached 900 Ω × cm^2^ [30]. Furthermore, the TJs in Caco-2 cells appear much tighter than those in the human intestine [55]. Thus, HIEC monolayers can be used for higher precision measurements compared with Caco-2 cell monolayers to predict compound absorption in humans [30,54]. In response to treatment with bLF, Caco-2 cells showed considerably enhanced barrier function compared with HIECs in this study; however, bLF showed a clear and similar ability to strengthen the barrier function of HIECs. Thus, HIECs may be a more suitable cell model than Caco-2 cells to reflect the barrier function of the intestinal epithelia.

## 4. Materials and Methods

### 4.1. Materials

bLF was purchased from Hilmar Ingredients (Dalhart, TX, USA); the purity of bLF was 96.2 ± 0.5% and its iron saturation was 12 ± 0.2%. Fetal bovine serum (FBS), Dulbecco’s modified Eagle’s medium (DMEM), Roswell Park Memorial Institute-1640 (RPMI-1640), Hank’s balanced salt solution (HBSS), and trypsin-EDTA were purchased from Gibco (Grand Island, NY, USA). Penicillin and streptomycin were purchased from Sigma-Aldrich (St. Louis, MO, USA). Cell Counting Kit-8 (CCK-8) was purchased from Dojindo Molecular Technologies, Inc. (Kumamoto, Japan). The Annexin V-FITC Apoptosis Detection Kit, Cell Cycle Analysis Kit, alkaline phosphatase (ALP) enzyme-linked immunosorbent assay (ELISA) kit, and bicinchoninic acid (BCA) protein assay kit were from Beyotime Institute of Biotechnology Co., Ltd. (Shanghai, China). The RNAprep Pure Cell Kit was purchased from Tiangen Biochemical Technology Co., Ltd. (Beijing, China). The PrimeScript^TM^ RT Reagent Kit and SYBR^®^ Premix Ex Taq^TM^ Kit were purchased from Takara Bio Ltd. (Shiga, Japan). Primary antibodies against β-actin, claudin-1, occludin, and ZO-1 were purchased from Abcam (Cambridge, UK), while secondary antibodies were purchased from Jackson ImmunoResearch Laboratories (West Grove, PA, USA). The protease inhibitor was purchased from EMD Millipore Corporation (Billerica, MA, USA) and phosphatase inhibitor was purchased from Roche (Mannheim, Germany). Ultrapure water was produced with the Milli-Q Plus system (EMD Millipore Corporation, Billerica, MA, USA). Other chemicals used in this study were of analytical grade.

### 4.2. Cell Lines and Culture Conditions

Caco-2 cells were kindly provided by the Stem Cell Bank, Chinese Academy of Sciences (Shanghai, China). Caco-2 cells were grown in DMEM supplemented with 20% (*v*/*v*) heat-inactivated FBS, 100 U/mL penicillin and streptomycin, and 2 mM L-glutamine. HIECs were purchased from the BeNa Culture Collection (Beijing, China) and HIECs were grown in RPMI-1640 supplemented with 10% (*v*/*v*) heat-inactivated FBS and 100 U/mL penicillin and streptomycin. The cells were maintained at 37 °C in an atmosphere of 5% CO2/95% air in a humidified incubator until they reached 80–90% confluence. The culture medium was changed every 1–2 days. The cells were subcultured by partial digestion with 0.25% trypsin incubated for 1–2 min or until the cells detached. Growth medium was then added to neutralize the trypsin. After centrifugation, the old medium was discarded, and fresh medium was added to re-suspend the cells. The cells were counted using trypan blue and the cell suspension was used to further culture the cells. Caco-2 cells from passages 42–50 and HIECs from passages 4–9 were used for all experiments.

### 4.3. Measurement of Cell Viability

Caco-2 cells were seeded into 96-well plates (10^4^ cells/well) and cultured for 3 day in DMEM with 5% FBS, and HIECs were seeded into 96-well plates (10^4^ cells/well) and cultured for 1 day with 3% FBS. After 1 day of serum starvation, the cells were exposed to five doses of bLF (10, 20, 50, 100, and 200 μg/mL) for 1, 2 and 3 days in FBS-free medium. After treatment, the old medium was discarded, and the cells were incubated with 100 μL CCK-8 solution (0.1 mg/mL in fresh medium) in an incubator at 37 °C for 2 h. The absorbance was read at a wavelength of 450 nm [56] (Bio-Rad Laboratories, Hercules, CA, USA). The cell viability is calculated as follows: Cell viability (%) = (OD_2_−OD_1_)/OD_1_ × 100%, where OD_2_ is the absorbance of sample groups at 450 nm, and OD_1_ is the absorbance of control groups at 450 nm.

### 4.4. Cell Cycle Analysis

For cell cycle analysis, 1 × 10^5^ cells seeded in 6-well multi-dishes at a total volume of 3 mL were incubated as described as 2.2. After 24 h of serum starvation, the cells were exposed to two different doses of bLF (50 and 100 μg/mL) for 48 h in FBS-free medium. Flow cytometric analyses were conducted using the FACS Calibur (BD Biosciences, Franklin Lakes, NJ, USA). At the end of incubation, the cells were rinsed twice with HBSS and trypsinized in trypsin-0.02% EDTA solution. After centrifugation for 5 min at 190× *g* at 4 °C, the supernatant was removed, and the pellet was resuspended in 1 mL of HBSS and centrifuged for 5 min at 190× *g*, after which the supernatant was discarded. Cold ethanol (75%, 2 mL) was added and the mixture was incubated at 4 °C overnight. After centrifugation for 5 min at 190× *g* and 4 °C, the pellet was treated with 10 μL RNase A and 25 μL propyl iodide staining solution at 37 °C for 30 min. The percentage of cells in each stage of the cell cycle was determined by counting 3 × 10^5^ cells using CellQuest software (BD Biosciences, San Diego, CA, USA).

### 4.5. Cell Differentiation Assay

Cell differentiation was assessed by measuring intestinal ALP activity according to the manufacturer’s instructions. Briefly, after the cells were treated with two different doses of bLF (50 and 100 μg/mL) for 7, 14, and 21 days, ALP activity was measured using the p-nitrophenyl phosphate (pNPP) method. Fifty microliters of the cell cytosol was incubated in a 96-well plate with 50 μL pNPP for 30 min at 37 °C, after which ALP activity was measured. The amount of liberated pNPP was determined spectrophotometrically at 405 nm using a microplate reader (Bio-Rad, Hercules, CA, USA).

### 4.6. Transepithelial Electrical Resistance (TEER) Measurement

The cells were seeded onto apical inserts (AP) (Millicell-24, Millipore, Burlington, MA, USA) at a density of 1 × 105 cells/insert (0.5 mL) and allowed to differentiate over the next 21 days. Medium was placed in the basolateral well (BL) (1.5 mL). TEER was determined with a Millicell-ERSR voltmeter (Millipore, Burlington, MA, USA) according to the manufacturer’s instructions. The assay was performed every 7 days during cells culture. Monolayers treated with DMEM alone were used as controls. The assay was performed in triplicate.

### 4.7. Determination of Epithelial Monolayer Permeability

Several chemicals such as mannitol, polyethylene glycol, dextran, and sodium fluorescein can be used to evaluate epithelial monolayer permeability [57]. At 21 days of cell culture, the growth media were replaced with HBSS for 2 h to facilitate protein/peptide depletion. For transport, HBSS in the apical chamber was replaced with 400 µL of 2 mg/mL sodium fluorescein, and fresh HBSS (600 µL) was added to the basolateral chamber. The concentrations of sodium fluorescein were measured in both the AP and BL layers with a microplate reader (Infinite M200 pro, Tecan, Männedorf, Switzerland) at 30, 60, and 120 min. The excitation and emission wavelengths were 495 nm and 520 nm, respectively. The *P*_app_ was used to evaluate the AP to BL paracellular permeability of the cell monolayers and calculated as follows: *P*_app_ = (dQ/dt)·(1/AC_0_), where dQ/dt is the permeability rate derived from the slope of the line, A is the surface area of the membrane, and C_0_ was the initial drug concentration in the AP compartment.

### 4.8. Immunofluorescence Analysis

The expression of TJ proteins was evaluated by immunofluorescence as previously described [58,59]. Confluent cell monolayers cultured on glass-bottom cell culture dishes were treated as described as 2.2, rinsed three times in HBSS, fixed with 4% paraformaldehyde at 4 °C for 20 min, and washed three times with HBSS. The cells were blocked with 5% FBS/HBSS for 1 h and then incubated with anti-claudin-1 (1:200), anti-occludin (1:200), or anti-ZO-1 antibody (1:100) overnight at 4 °C. This was followed by incubation with secondary antibody anti-rabbit IgG (1:200) at 37 °C for 30 min. After rinsing with HBSS three times, the cells were mixed with 4′,6-diamidino-2-phenylindole and incubated at 20 °C for 10 min. Following extensive rinsing, images were captured by laser scanning fluorescence microscopy (AxioScope.A 1, Carl Zeiss AG, Oberkochen, Germany).

### 4.9. Real-Time Quantitative Polymerase Chain Reaction (qRT-PCR) Analysis

Total RNA was extracted using TRIzol reagent according to the manufacturer’s instructions. The purity and concentration of RNA were determined with a NanoDrop spectrophotometer (Thermo Fisher, Waltham, MA, USA). One microgram of total RNA was reverse-transcribed into cDNA using the PrimeScript RT Reagent Kit, and qRT-PCR was performed in 20 μL reactions for each candidate gene and *GAPDH* (internal control) using the SYBR^®^ Premix Ex Taq^TM^ Kit. Relative mRNA expression levels were calculated based on Ct values [60]. The geometric mean expression of *GAPDH* was used for normalization. All primer pairs (Table 1) were synthesized by Sangon Biotech (Shanghai, China).

### 4.10. WB Analysis

TJ proteins in the cell monolayers were measured as described previously [61]. Briefly, confluent Caco-2 cells or HIECs were treated with or without bLF for 48 h. Total protein was harvested from the cells in each well using 100 μL of ice-cold lysis buffer supplemented with a protease inhibitor and phosphatase inhibitor. A BCA protein assay kit was used to quantify the protein content. Protein samples were separated by SDS-PAGE and transferred to nitrocellulose membranes for up to 1 h. The membranes were blocked in Tris-buffered saline containing 0.1% Tween 20 with 5% skim milk for 1 h at 20 °C. Next, the membranes were incubated with primary antibodies claudin-1 (1:1000), occludin (1:1000), or ZO-1 (1:1000) overnight at 4 °C. The following day, the membranes were incubated with secondary antibodies for 1 h at 20 °C. Protein bands were visualized on X-ray film. Image J software was used to analyze gray level (NIH, Bethesda, MD, USA), and the target protein/β-actin ratio was used to represent the relative expression of the target protein.

### 4.11. Statistical Analysis

All reported data were collected from three independent assays. The results are expressed as the mean values or mean values ± standard deviations. The data were analyzed using the Statistix 8.1 software package (Analytical Software, St. Paul, MN, USA). Significant differences (*p* < 0.05) among the means were identified by the Tukey test.

## 5. Conclusions

In conclusion, the present data indicate that bLF increases the expression of TJ proteins, and protects the intestinal epithelial barrier function. bLF initially exerted a marked effect on intestinal epithelial cell growth and then on cell differentiation. Meanwhile, bLF enhanced the integrity of the intestinal epithelial barrier by increasing TEER values, decreasing *P*_app_ values, and up-regulating TJ protein expression at both protein and mRNA level. HIECs are useful for analyzing the regulation of proliferation, integrity, and expression of TJ proteins in the intestinal epithelial barrier, and thus, would be an ideal model for future studies of intestinal barrier function. The molecular mechanism responsible for bLF-mediated intestinal barrier integrity has been preliminarily verified, however, further studies would be required to confirm these findings fully in vivo cases.

## Figures and Tables

**Figure 1 molecules-24-00148-f001:**
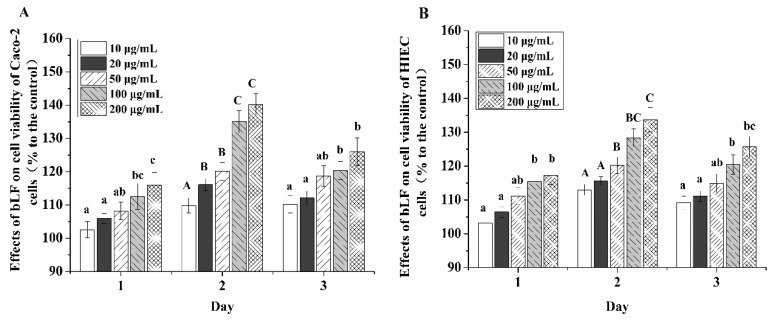
Cell viability values of Caco-2 (**A**) and HIEC cells (**B**) incubated with bLF at two different doses for 1, 2, and 3 days. A–C, a–c different letters above the bars indicate significant differences (*p* < 0.05).

**Figure 2 molecules-24-00148-f002:**
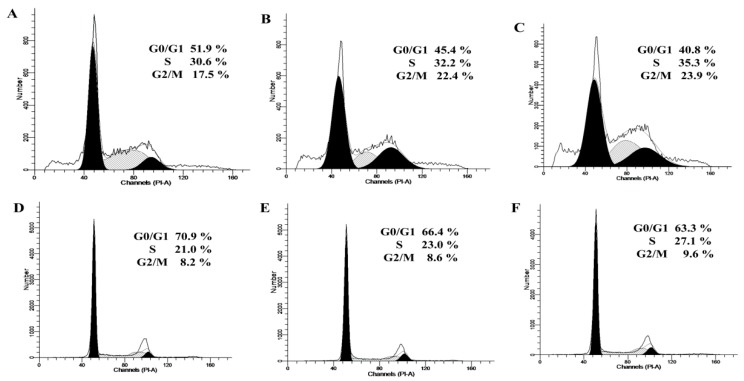
Cell-cycle distribution of Caco-2 cells treated without (**A**) or with bLF at two different doses (**B**,**C**) for 48 h, or HIEC cells treated without (**D**) or with bLF at two different doses (**E**,**F**) for 48 h.

**Figure 3 molecules-24-00148-f003:**
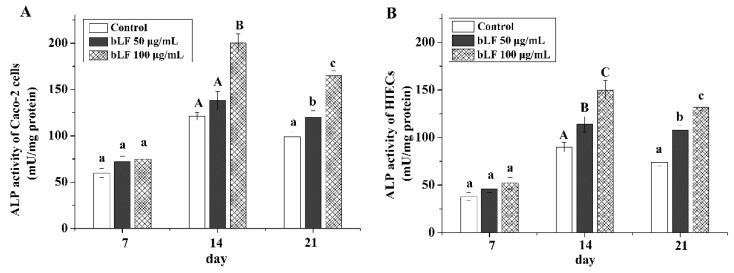
ALP acticity of Caco-2 cells (**A**) and HIECs (**B**) incubated with bLF at two different doses for 7, 14, and 21 days. A–C, a–c different letters above the bars indicate significant differences (*p* < 0.05).

**Figure 4 molecules-24-00148-f004:**
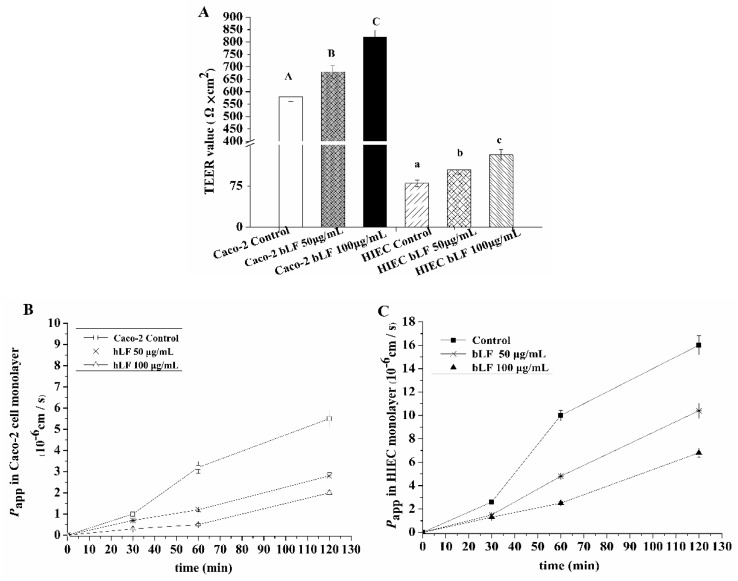
TEER values (**A**) of Caco-2 and HIEC cells, and their *P*_app_ values ((**B**), Caco-2 cells; (**C**), HIEC cells). A–C, a–c different letters above the bars indicate significant differences (*p* < 0.05).

**Figure 5 molecules-24-00148-f005:**
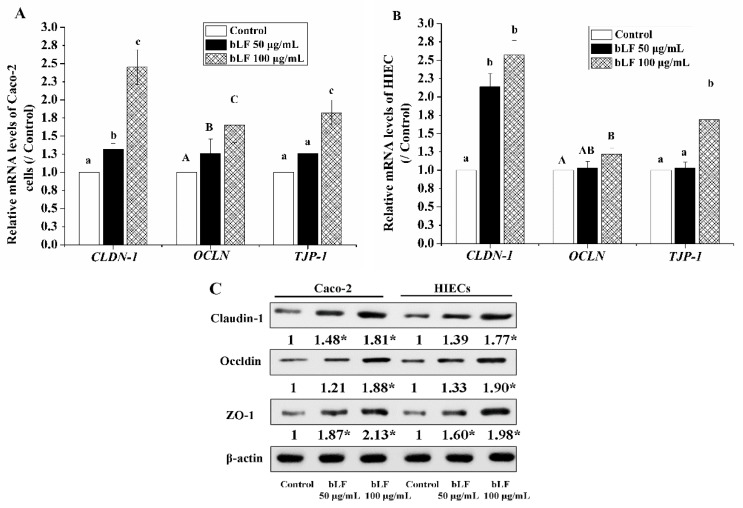
bLF improved the mRNA of Caco-2(**A**) and HIEC cells (**B**) and protein expressions of TJ proteins (**C**). * indicate difference from the control group (*p* < 0.05).

**Figure 6 molecules-24-00148-f006:**
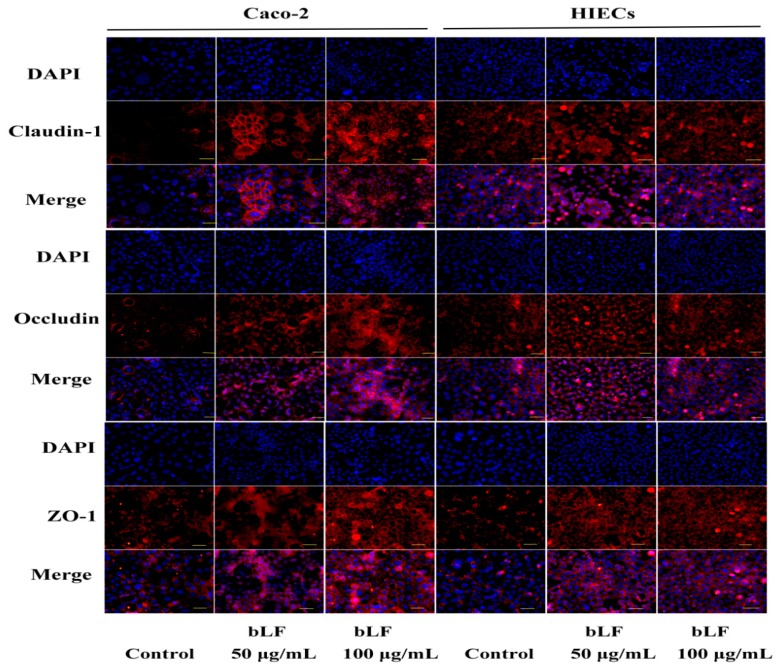
bLF improved morphological disruption of TJ proteins after 48 h incubation. Scale bar: 100 μm.

**Table 1 molecules-24-00148-t001:** PCR primer sequences.

Gene Name	NCBI Reference Sequence	Forward Primer (5′-3′)	Reverse Primer (5′-3′)
*CLDN-1*	9076	GCGACAACATCGTGACCG	CCAACCACCATCAAGGCAC
*OCLN*	100506658	CCCCATCTGACTATGTGGAAAG	CAGGCGAAGTTAATGGAAGC
*TJP-1*	7082	GAGTGAACCACGAGACGCTG	TTCCGAGATTCTGGACATAACC
*ACTB*	60	TGACGTGGACATCCGCAAAG	CTGGAAGGTGGACAGCGAGG

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
