# Peer review of "The In Vitro Protective Role of Bovine Lactoferrin on Intestinal Epithelial Barrier"

_molecules, 2019, doi:10.3390/molecules24010148_

Round 1
Reviewer 1 Report
The manuscript “Protective Role of Bovine Lactoferrin on Intestinal Epithelial Barrier in Vitro” show interesting research issues. This kind of research can contribute to the development of nutraceutical products that support disease prevention or treatment. It was found that the bovine lactoferrin may enhance intestinal epithelial barrier function of HIECs by improving the expression of selected proteins. The sentence concerning the aim (L. 80-82) should be schowed as a hypothesis (please change).
Specific comments
Abstract
L.16. Abstract should not contain elements of discussion.
L.29-30. bLF does not apply directly to functional foods. Sentence is not true. Explanation: Functional food is consumed as part of a normal food pattern. It is not a pill, a capsule or any form of dietary supplement.
Introduction
This chapter is good developed.
L.78-92. There is no central hypothesis.
2. Materials and Methods
The research area and methods are acceptable. But there are some understatement.
2.10. Statistical analysis
The distribution of variables is normal?
4. Discussion
L.304. 2 g/kg ? What applies DM or BM?
The discussion section should be providing concise background relevant to this study. Please indicate the significance.
Assess the impact of factors on the efficiency of the intestinal barrier can be made to work:Anderson, R. C., MacGibbon, A. K., Haggarty, N., Armstrong, K. M., & Roy, N. C. (2018). Bovine dairy complex lipids improve in vitro measures of small intestinal epithelial barrier integrity. PloS one, 13(1), e0190839.
5. Conclusions
L.345-346 please transfer to 351.
L. 352. Please define what to refer to further studies.
Author Response
Thank you for giving us an opportunity to revise our manuscript. We appreciate you very much for your constructive comments and suggestions on our manuscript (Molecules-410639)
We have studied the comments carefully and have made the corresponding corrections, which we hope meet with approval. We have moved Part 2 (Materials and Methods) of the original manuscript to Part 4 in the revised manuscript according to the template of Molecules, and re-arranged the reference. We did not list all of the changes but marked in the revised paper with highlighted in yellow. In addition, our manuscript has been checked by editors at Editage, a professional English editing service company. Attached please find the Certificate of English Editing. The following pages detail our answers to the questions.
Merry X’Mas!
Thank you and best regards.
Sincerely,
Xiao-Xi Xu
College of Food Science, Northeast Agricultural University
Harbin 150030, PR China
Tel: +86-451-55190459.
Fax: +86-451-55190577;
E-mail: [email protected];
Comments and Suggestions for Authors
The manuscript “Protective Role of Bovine Lactoferrin on Intestinal Epithelial Barrier in Vitro” show interesting research issues. This kind of research can contribute to the development of nutraceutical products that support disease prevention or treatment. It was found that the bovine lactoferrin may enhance intestinal epithelial barrier function of HIECs by improving the expression of selected proteins. The sentence concerning the aim (L. 80-82) should be schowed as a hypothesis (please change).
Response: Thanks very much for your helpful comments and suggestions. We had tried our best to improve the manuscript and made some changes in the original manuscript which we hope meet with approval.
The statement of “The aim of this study was to provide a better understanding of the bioactivities of bLF in the intestinal barrier function, and to verify if HIEC could be used as a cell model in human intestinal epithelial barrier function study.” in the original manuscript was corrected in the revised manuscript as “We hypothesized that bLF would play an beneficial role on the intestinal epithelial barrier and HIEC would be an ideal cell mode to be used in human intestinal epithelial barrier function study”.
Please check in Page 2, Line 88-90.
Specific comments
Abstract
Point 1: L.16. Abstract should not contain elements of discussion.
Response: We are very sorry for our incorrect writing about Abstract in Line 16. “Especially on human intestinal epithelial crypt cells (HIECs), although recent reports suggested that HIECs were more similar to those in the human small intestine” in the original manuscript was changed into “Human intestinal epithelial crypt cells (HIECs) were more similar to those in the human small intestine, compared with the well-established Caco-2 cells” in the revised paper as you suggested.
Please check in Page 1, Line 15-17.
Point 2: L.29-30. bLF does not apply directly to functional foods. Sentence is not true. Explanation: Functional food is consumed as part of a normal food pattern. It is not a pill, a capsule or any form of dietary supplement.
Response: Thank you for the helpful comment. We are sorry for this mistake. The sentence in the original manuscript “Therefore, bLF highlights the potential benefits of using as functional foods to inflammatory bowel diseases which are caused by the loss of barrier integrity.” were changed in the revised paper into “Therefore, bLF may be incorporated into functional foods for treatment of inflammatory bowel diseases which are caused by loss of barrier integrity.”
Please check in Page 1, Line 26-28.
Introduction
This chapter is good developed.
Point 3: L.78-92. There is no central hypothesis.
Response: Thank you so much for your valuable comments. We have made correction according to your good comments.
Please check in Page 2, Line 86-90.
2. Materials and Methods
Point 4: The research area and methods are acceptable. But there are some understatement.
Response: We have re-written this part as you suggested, and moved this part from part 2 in the original manuscript in to part 4 in the revised paper according to the template of this journal. We made some modifications and details of methods were added. We marked these changes in the revised paper with highlighted in yellow.
Please check in Page 8, Line 242-243.
Please check in Page 9, Line 251-255, Line 258-259, Line 262, Line266, Line 269-271, Line275, Line 278-279, Line 281-284, Line 285-289.
Please check in Page 10, Line 293, Line 295-296, Line 297, Line 299, Line 301-306, Line 308-309, Line 311, Line 313-315, Line 317, Line 320-321, Line 323-329, Line 332, and Line 334, and Line 336.
Please check in Page 11, Line 338-339, Line 341, Line 343, Line 344-346, Line 350, Line 352, Line 355, Line 357, and Line 359.
Thank you.
2.10. Statistical analysis
Point 5: The distribution of variables is normal?
Response: We analyzed these data with the software Statistix to determine the data between the different groups had significant differences. Very thanks.
4. Discussion
Point 6: L.304. 2 g/kg ? What applies DM or BM?
Response: Thank you for your remind. We are sorry for had given an unclear expression here.
We had revised the sentence as this reported study used ZnO at 2 g/kg for the basal diet.
Please check in Page 8, Line 201.
Point 7: The discussion section should be providing concise background relevant to this study. Please indicate the significance.
Response: Thank you for your valuable comment.
Firstly, in the discussion we emphasized the important of intestinal barrier integrity, and stated the previous studies on the critical protective effect of TJ proteins to the intestinal barrier function aimed to provide support for our present study.
Secondly, we paid attention to the beneficial role of dietary components on intestinal barrier, which are reasonable theoretical support for bLF has the ability to improve the barrier function by increasing the expression of TJ proteins. In this part, we added contents relevant to the mechanism of bLF performed on the two cells.
Please check in Page 8, Line 212-225.
Last but not the least; we illustrated the characteristic and recent studies of Caco-2 and HIECs cells, which suggested the significance of the two cells used in the present study.
Point 8: Assess the impact of factors on the efficiency of the intestinal barrier can be made to work: Anderson, R. C., MacGibbon, A. K., Haggarty, N., Armstrong, K. M., & Roy, N. C. (2018). Bovine dairy complex lipids improve in vitro measures of small intestinal epithelial barrier integrity. PloS one, 13(1), e0190839.
Response: Thank you for your suggestion. We have cited this paper in our revised manuscript.
Please check in Page 10, Line 325 and the Reference part (Page 14, Line 519-520).
5. Conclusions
Point 9: L.345-346 please transfer to 351.
Response: We have made this modification as your suggestion.
Please check in Page 11, Line367--369.
Very thanks.
Point 10: L. 352. Please define what to refer to further studies.
Response: Thank you for you helpful comments. The “further studies” here refers to “studies in vivo”. The statement “further studies are needed to confirm these findings fully” were modified to “further studies would be required to confirm these findings fully in vivo cases”.
Please check in Page 11, Line 369-371.

Reviewer 2 Report
The manuscript by Zhao et al. demonstrates the possible function of bLF to tight junction using two types of cells, Caco-2 and HIEC. The authors revealed that bLF showed higher activity in Caco-2, and also, showed desired responses to barrier function in HIECs. Caco-2 cells are cancer cells. On the other hand, HIECs are normal cells. But, the authors did not discuss the point concerning cell types. And, the data did not show the precise results and the layout is not good in all figures. And also, the manuscript in English is poor. Thus, I will recommend to answer the following major concerns.
Abstract
Page1, Line29~: "Therefore, bLF" is in bold face.
Introduction
Page1, Line42~: The authors quote Ref.8 for suggestion. There is no respect for the authors in Ref. 8. And also, LF is a bovine origin?
Page2, Line78-82: Recently, intestinal organoid culture is normally used for the research. Could you tell me the reason why you used two types of intestinal epithelial cells? One is a cancer cell, the other is a normal cell.
Materials and Methods
Page2, Line93: Antibody against claudin is claudin1?
Page3, Line2: The authors did not show phosphate inhibitor perchased.
Page3, Line101-115: The same content is included in 2.1. Please rewrite.
Page3, Line131: Cold alcohol (75%, 2mL) is vodka? Cold alcohol is cold ehtanol?
Page3, Line 136: Did you examine the effect of bLF on cell differentiation with another method(s)?
Page4, Line166: The authors washed three times. with what?
Page4, Line 167: The authors incubated with anti-claudin. Anti-claudin-1
Page4, Line172: The authors showed QT-PCR. But, in the manuscript, the authors showed qRT-PCR (Line176).
Table1: Gene name is italic. NCBI reference sequence is correct?
Results
Page5, Line 203-204: The authors concluded that HIECs were less sensitive than Caco-2 cells to bLF. Why did the authors use the cells?
Page 5, (Figure 1): I cannot understand the figure. Where is control bar graph? At the white bar graph, there is a significant difference. I will reject the figure.
Page6, (Figure2): How many time did you carry out the experiment?
Page6: (Figure 3): Please separate the data between Caco-2 and HIECs.
Page7, Line263: Why did you decide to analyze the expression of CLDN-1?
Page7, (Figure5A): Please separate the data between Caco-2 and HIECs.
Page7, (Figure5A): What is the line in Figure 5A and 5B?
Page8, (Figure 6): In Figure 5B, the authors showed WB result. I think there is no big difference between treated and untreated control. The pictures of Figure 6 is bad. Please show the enlargements.
Page8, Line285-286: The authors concluded that the results directly proved that bLF had the ability to increase TJ proteins production and thereby strengthen barrier function. I do not think so.
Discussion
1) Molecular weight of bLF is about 83Dka. I wonder that the molecule penetrate into the cells or bind cell surface. Please give me the answer.
2) Please tell me the significance in the two cell types used in this manuscript.
3) What kind of signaling pathway(s) is enhanced by bLF?
Page8, Line299: The authors told that "these results support those of the present study". Please tell me how the references support the present study
Page9, Line311: the intestinal epithelia -> the intestinal epithelium.
Page9, Line315-316: The authors told that HIECs are not widely used as a cell model in studies of intestinal TJs. Why did the authors used the cell line?
Page9, Line 326-338: The sentences should be moved to" the introduction" for readers.
Author Response
Thank you for giving us an opportunity to revise our manuscript. We appreciate you very much for your constructive comments and suggestions on our manuscript (Molecules-410639)
We have studied the comments carefully and have made the corresponding corrections, which we hope meet with approval. We have moved Part 2 (Materials and Methods) of the original manuscript to Part 4 in the revised manuscript according to the template of Molecules, and re-arranged the reference. We did not list all of the changes but marked in the revised paper with highlighted in yellow. In addition, our manuscript has been checked by editors at Editage, a professional English editing service company. Attached please find the Certificate of English Editing. The following pages detail our answers to the questions.
Merry X’Mas!
Thank you and best regards.
Sincerely,
Xiao-Xi Xu
College of Food Science, Northeast Agricultural University
Harbin 150030, PR China
Tel: +86-451-55190459.
Fax: +86-451-55190577;
E-mail: [email protected];
Comments and Suggestions for Authors
The manuscript by Zhao et al. demonstrates the possible function of bLF to tight junction using two types of cells, Caco-2 and HIEC. The authors revealed that bLF showed higher activity in Caco-2, and also, showed desired responses to barrier function in HIECs. Caco-2 cells are cancer cells. On the other hand, HIECs are normal cells. But, the authors did not discuss the point concerning cell types. And, the data did not show the precise results and the layout is not good in all figures. And also, the manuscript in English is poor. Thus, I will recommend to answer the following major concerns.
Thanks very much for your valuable comments and suggestions. We had tried our best to improve the manuscript and made some changes in the original manuscript which we hope meet with approval. All of the figures were redesigned. In addition, our manuscript has been checked by editors at Editage, a professional English editing service company. We did not list all of the changes but marked in the revised paper with highlighted in yellow.
Thanks again.
Abstract
Point 1: Page1, Line29~: "Therefore, bLF" is in bold face.
Response: We are very sorry for our negligence of this character font. The bold "Therefore, bLF" was corrected to Palatino Linotype "Therefore, bLF".
Please check in Page 1, Line 26.
Thank you.
Introduction
Point 2: Page1, Line42~: The authors quote Ref.8 for suggestion. There is no respect for the authors in Ref. 8. And also, LF is a bovine origin?
Response: Thank you for your comments. We are sorry for have not given much respect for the authors in Ref.8. LF used in Ref. 8 is human origin, and LF in Ref.7 is bovine origin. We add the detail of LF in the revised manuscript.
Please check in Page 1, Line 40-43.
Point 3: Page2, Line78-82: Recently, intestinal organoid culture is normally used for the research. Could you tell me the reason why you used two types of intestinal epithelial cells? One is a cancer cell, the other is a normal cell.
Response: Thank you for your valuable comments.
(1) We used two cell lines in this study, aimed to compare the difference response of Caco-2 and HIECs to bLF stimulate.
(2) Caco-2 is a human cancer cell line and has been used to investigate various aspects of intestinal cell function and regulation. Please see:
1. Lactoferrin induces concentration-dependent functional modulation of intestinal proliferation and differentiation. Pediatric Research. 2007, 61,410-414.
2. Bovine lactoferrin can be taken up by the human intestinal lactoferrin receptor and exert bioactivities. Journal of Pediatric Gastroenterology & Nutrition. 2011, 53, 606-614.
3. Effects of lactoferrin on intestinal epithelial cell growth and differentiation: an in vivo and in vitro study. Biometals. 2014, 27, 857-874.
4. Resveratrol ameliorates intestinal barrier defects and inflammation in colitic mice and intestinal cells. Journal of agricultural and food chemistry. 2018. Doi:10.1021/acs.jafc.8b04138
5. The function and mechanism of preactivated thiomers in triggering epithelial tight junctions opening. European Journal of Pharmaceutics and Biopharmaceutics. 2018, 133, 188-199.
(3) HIECs had been rarely used in the study of intestinal barrier function. Recent study suggested that HIECs is similar to those in the human small intestine (Application of a human intestinal epithelial cell monolayer to the prediction of oral drug absorption in humans as a superior alternative to the Caco-2 cell monolayer. J Pharm Sci-US. 2016, 105, 915-924.; Human milk exosomes and their microRNAs survive digestion in vitro and are taken up by human intestinal cells. Mol Nutr Food Res. 2017, 61, DOI:10.1002/mnfr.201700082.).
Above all, we used both Caco-2 cells and HIECs in the present study.
Materials and Methods
Point 4: Page2, Line93: Antibody against claudin is claudin1?
Response: We are sorry for have not written this clearly. The antibody is claudin-1, it was purchased from Abcam, and the product No. is ab 15098. We modified this mistake in the revised manuscript.
Please check in Page 9, Line 255.
Point 5: Page3, Line2: The authors did not show phosphate inhibitor perchased.
Response: Thank you for your remind. The purchase information of phosphate inhibitor was added in the manuscript.
Please check in Page 9, Line 258-259.
Point 6: Page3, Line101-115: The same content is included in 2.1. Please rewrite.
Response: Thank you for this reminding. We have re-written this part.
Please check in Page 9, Line 265-270.
Point 7: Page3, Line131: Cold alcohol (75%, 2mL) is vodka? Cold alcohol is cold ehtanol?
Response: Thank you for your remind. We had made some pen slips in the previous manuscript. It should be cold ethanol here, and we corrected that in the revised manuscript.
Please check in Page 10, Line 293.
Point 8: Page3, Line 136: Did you examine the effect of bLF on cell differentiation with another method(s)?
Response: Thank you for your valuable comments. Overall, ALP activity is regarded as classic index to reflect cell differentiation, and has been used in many studies. Please see these published papers.
1. Noda S.; Yamada A.; Nakaoka K.; Goseki-Sone M.1-alpha,25-Dihydroxyvitamin D-3 up -regulates the expression of 2 types of human intestinal alkaline phosphatase alternative splicing variants in Caco-2 cells and may be an important regulator of their expression in gut homeostasis. Nutr Res. 2017, 46, 59-67.
2. Ferraretto A.; Bottani M.; De L. P.; Cornaghi L.; Arnaboldi F.; Maggioni M.; Amelia F.; Elena D. Morphofunctional properties of a differentiated Caco2/HT-29 co-culture as an in vitro model of human intestinal epithelium. Biosci Rep. 2018, 38, 2.
3. Huang J. Y.; Zhang L.; Chen P.; Chen S.; Wu Y.; Tang S. Q. Transportation efficiency of insulin loaded in agarose-grafting-hyaluronan microparticle crossing Caco-2 cell monolayer. Curr Appl Phys. 2011, 11, 794-799.
Thus, we also used this assay in the present work.
Point 9: Page4, Line166: The authors washed three times. with what?
Response: Thank you for your remind. It should be that washed three times with the HBSS (Hank's balanced salt solution), and we have added this content in the revised manuscript.
Please check in Page 10, Line 327.
Point 10: Page4, Line 167: The authors incubated with anti-claudin. Anti-claudin-1?
Response: We are sorry for our incorrect writing here. The antibody should be anti-claudin-1. We changed “anti-claudin” to “anti-claudin-1” in the revised manuscript.
Please check in and Page 10, Line 328.
Point 11: Page4, Line172: The authors showed QT-PCR. But, in the manuscript, the authors showed qRT-PCR (Line176).
Response: Thank you for your remind. We modified it into qRT-PCR uniformly in the revised paper.
Please check in Page 10, Line 334.
Point 12: Table1: Gene name is italic. NCBI reference sequence is correct?
Response: We are sorry for the negligence of this character font. We changed the gene name into italic as you suggested, and confirmed the reference sequence with the information from NCBI website.
Please check in Page 11, Line 343.
We also had changed gene name into italic in the Results part.
Please check in Page 5, Line 157-160.
Thank you very much.
Results
Point 13: Page5, Line 203-204: The authors concluded that HIECs were less sensitive than Caco-2 cells to bLF. Why did the authors use the cells?
Response: Thank you for bringing this to our attention. We investigated the performance of these two cell lines in order to predict human intestine response to bLF. HIECs were less sensitive than Caco-2 in this study, but do not mean HIECs were bad. The data were more closer to human response more better. And recent studies reported that although the TEER value of HIEC monolayers was lower than Caco-2, but it is close to that in the human small intestine (Application of a human intestinal epithelial cell monolayer to the prediction of oral drug absorption in humans as a superior alternative to the Caco-2 cell monolayer. J Pharm Sci-US. 2016, 105, 915-924.). Furthmore, the TJs in Caco-2 cells appeared much tighter than those in human intestine (Cell culture-based models for intestinal permeability: a critique. Drug Discov Today. 2005, 10, 335-343.). So we used HIECs in our present study.
Point 14: Page 5, (Figure 1): I cannot understand the figure. Where is control bar graph? At the white bar graph, there is a significant difference. I will reject the figure.
Response: Thank you for your helpful comments. The data of control groups were defined as 100%, so we did not showed control bar on Figure 1. We had added the calculation of cell viability in the revised manuscript.
Please check in Page 9, Line 282-284.
And the significance markers of Figure 1 were modified in order to read more clearly.
Please check in Page 3, Line 102-105.
Point 15: Page6, (Figure2): How many time did you carry out the experiment?
Response: Thanks for your helpful comments. We carried out three independent cell culture experiments with each condition, and in each independent cell culture system, we took 3 samples to determine the distribution of cell-cycle phase. We showed the results by mean values. Usually, the results of cell-cycle distribution were showed by mean values. Please see:
1. Ethanol extract of abnormal savda munziq, a herbal preparation of traditional uighur medicine, inhibits caco-2 cells proliferation via cell cycle arrest and apoptosis. Evid-Based Compl Alt. 2012, 2012, 1-6.
2. Mechanism of alternariol monomethyl ether-induced mitochondrial apoptosis in human colon carcinoma cells. Toxicology. 2011, 290, 231-241.
3. Synthesis, characterization and biological evaluation of some new indomethacin analogs with a colon tumor cell growth inhibitory activity. Med Chem Res. 2017, 26, 2205-2220.
Point 16: Page6: (Figure 3): Please separate the data between Caco-2 and HIECs.
Response: Thank you for your helpful suggestion. We modified Figure 3 as you suggested.
Please check in Page 4, Line 131.
Point 17: Page7, Line263: Why did you decide to analyze the expression of CLDN-1?
Response: The protein encoded by this gene, a member of the claudin family, is an integral membrane protein and a component of tight junction strands (please see the information form the NCBI gene bank).Please using the link below.
https://www.ncbi.nlm.nih.gov/gene/9076
Point 18: Page7, (Figure5A): Please separate the data between Caco-2 and HIECs.
Response: Thanks for your suggestion. We modified Figure 5A as you suggested.
Please check in Page6, Figure 5A and 5B.
Point 19: Page7, (Figure5A): What is the line in Figure 5A and 5B?
Response: We are very sorry for this mistake. It was the word typesetting problem. We corrected this mistake in the revised manuscript.
Please check in Page6, Figure 5.
Point 20: Page8, (Figure 6): In Figure 5B, the authors showed WB result. I think there is no big difference between treated and untreated control. The pictures of Figure 6 is bad. Please show the enlargements.
Response: Thank you for your valuable suggestion and comment. Some of the WB results between treated and untreated control were not significant, but some of them were significant difference, and we marked “*” on the data.
We modified Figure 6 with higher dpi pictures and enlarged them.
Please check in Page 7, Figure 6.
Point 21: Page8, Line285-286: The authors concluded that the results directly proved that bLF had the ability to increase TJ proteins production and thereby strengthen barrier function. I do not think so.
Response: Thank you very much for your valuable comments.
(1) The present study showed that bLF increased TEER values, decreased Papp values, and up-regulated TJ protein expression at both protein and mRNA level.
(2) Previous studies demonstrated the intestinal barrier is thought to greatly depend on the formation of specialized intercellular TJs. Up-regulating the expression of TJ proteins, the intestinal barrier function would be improved. Please see follows:
1. Lam Y.Y.; Ha C.W.Y.; Hoffmann J.M.A.; Oscarsson J.; Dinudom A.; Mather T.J.; Cook D.I.; Hunt N.H.; Caterson I.D.; Holmes A.J.; Storlien L.H. Effects of dietary fat profile on gut permeability and microbiota and their relationships with metabolic changes in mice. Obesity. 2015, 23, 1429-1439.
2. Santos P.S.; Caria C.R.P.; Gotardo E.M.F.; Ribeiro M.L.; Pedrazzoli J.J.; Gambero A. Artificial sweetener saccharin disrupts intestinal epithelial cells' barrier function in vitro. Food Funct. 2018, 9, 3815-3822.
3. Hu C.H.; Xiao K.; Song J.; Luan Z.S. Effects of zinc oxide supported on zeolite on growth performance, intestinal microflora and permeability, and cytokines expression of weaned pigs. Anim Feed Sci Tech. 2013, 181, 65-71.
4. Li Y.; Wang X.; Li N.; Li J. The study of n-3PUFAs protecting the intestinal barrier in rat HS/R model. Lipids Health Dis. 2014, 13, 146.
5. Song P.; Zhang R.; Wang X.; He P.; Tan L.; Ma X. Dietary grape-seed procyanidins decreased postweaning diarrhea by modulating intestinal permeability and suppressing oxidative stress in rats. J Agr Food Chem. 2011, 59, 6227-6232.
(3) So we conclude that bLF had the ability to increase TJ proteins production and thereby strengthen barrier function.
Discussion
Point 22: 1) Molecular weight of bLF is about 83Dka. I wonder that the molecule penetrate into the cells or bind cell surface. Please give me the answer.
Response: Thank you very much for this helpful comment.
(1) Previous studies investigated the response of many cell lines to bLF, such as Caco-2 cells (The impact of lactoferrin with different levels of metal saturation on the intestinal epithelial barrier function and mucosal inflammation. Biometals. 2016, 29, 1019-1033.), Hs578T breast cancer cells (In vitro evaluation of bovine lactoferrin potential as an anticancer agent. International Dairy Journal. 2015, 40, 6-15.), and THP1 macrophages cells (Effect of lactoferrin protein on red blood cells and macrophages: mechanism of parasite-host interaction. Drug Design, Development and Therapy. 2015, 9, 3821-3835.). The results of those studies demonstrated that bLF could affect the cells and exert biofunctions.
(2) We did not focus on the issue that whether bLF penetrated into the cells or bind cell surface in the present study. But the previous studies illustrated that bLF exerted its multiple biological activities by binding to its receptor on the cell membrane and then activate signaling transduction (Apo-and holo-lactoferrin stimulate proliferation of mouse crypt cells but through different cellular signaling pathways. Int J Biochem Cell B. 2012, 44, 91-100.). bLF is resistant to proteolytic digestion, particularly when the pH is not very low, such as in infants (The effect of trypsin on bovine transferrin and lactoferrin. BBA. 1976, 446, 214-225.), however, the resitant bLF could be absorbed in blood and various tissues of mouse, rats and piglets (Uptake of ingested bovine lactoferrin and its accumulation in adult mouse tissues. Int Immunopharmac. 2007, 7, 1387-1393.; Evidence of lactoferrin transportation into blood circulation from intestine via lymphatic pathway in adult rats. Exp Physiol. 2004, 89, 263-270.; Characteristic transport of lactoferrin from the intestinal lumen into the bile via the blood in piglets. Comp Biochem and Phys A. 1999, 124, 321-327.).
Point 23: 2) Please tell me the significance in the two cell types used in this manuscript.
Response: Thank you so much for you valuable comment.
We used two cell lines in this study, one was colonic origin Caco-2, the other one was HIEC. Caco-2 is one of the human colon cancer cell lines, have been used as gold standard cell model to investigate various aspects of intestinal cell function and regulation. Nevertheless, this cell model has different performance compared with the human small intestine, such as higher TEER values. Therefore, it is not possible to full predict the behavior and prospects of human intestine. So, we used HIECs, which is normal cell model origin from human fetal. HIECs had been rarely used in the study of intestinal barrier function. We aimed to compare the difference response of Caco-2 and HIECs to bLF stimulate and to verify if HIECs is an ideal model in the study of intestinal barrier function.
Point 24: 3) What kind of signaling pathway(s) is enhanced by bLF?
Response: A good comment!
We thought bLF might play beneficial role on the intestinal epithelium barrier through NK-κB signal pathway. bLF could regulate the inflammatory and immune system. TNF-α is one of the important activator of NF-κB signaling pathway. bLF significantly decreased the mRNA expression of TNF-α in a dose-dependent manner in our study, but which needs to be investigate in the further experiment, so data were not show in this manuscript.
In the revised Discussion section, we added sentences to show the potential signal pathway in the Discussion part.
Please check in Page 8, Line 220-225.
Very thank!
Point 25: Page8, Line299: The authors told that "these results support those of the present study". Please tell me how the references support the present study
Response: Thank you for this helpful comment. The intestinal barrier is thought to greatly depend on the formation of specialized intercellular TJs. In the present study, bLF up-regulated the expression of TJ protein, therefore the intestinal barrier function was improved, just as the results in the previous studies. We changed “these results support those of the present study” into “These results support those of the present study that up-regulate the expression of TJ protein can improve the intestinal barrier function”.
Please check in Page 8, Line 195-196.
Point 26: Page9, Line311: the intestinal epithelia -> the intestinal epithelium.
Response: We have made correction according your comments.
Please check in Page 8, Line 227.
Point 27: Page9, Line315-316: The authors told that HIECs are not widely used as a cell model in studies of intestinal TJs. Why did the authors used the cell line?
Response: Thank you for your valuable comment. Below is our reply
(1) HIECs are used in the study of intestinal permeability recently (please see: Application of a human intestinal epithelial cell monolayer to the prediction of oral drug absorption in humans as a superior alternative to the Caco-2 cell monolayer. J Pharm Sci-US. 2016, 105, 915-924; and A pumpless body-on-a-chip model using a primary culture of human intestinal cells and a 3D culture of liver cells. Lab Chip. 2018, 18, 2036-2046.)
(2) Another study also indicated that HIECs are promising tools for the study of intestinal barrier function. (please see: Human cell models to study small intestinal functions: Recapitulation of the crypt-villus axis. Microsc Res Techni. 2000, 49, 394-406.)
(3) Most importantly, recent study suggested that HIECs is similar to those in the human small intestine (Application of a human intestinal epithelial cell monolayer to the prediction of oral drug absorption in humans as a superior alternative to the Caco-2 cell monolayer. J Pharm Sci-US. 2016, 105, 915-924.; Human milk exosomes and their microRNAs survive digestion in vitro and are taken up by human intestinal cells. Mol Nutr Food Res. 2017, 61, DOI:10.1002/mnfr.201700082.).
Based on these studies, we though HIECs might be an ideal cell line in the study of intestinal barrier function, so we used this cell line in the present study.
Thank you again.
Point 28: Page9, Line 326-338: The sentences should be moved to" the introduction" for readers.
Response: Thank you for your helpful comments. We moved the sentences of Line 326-338 in the original paper to “the introduction” in the revised paper.
Please check in Page 2, Line 58-68.
We also rearranged the reference that corresponding to these contents.
Please check in Page 13-14, Reference.

Round 2
Reviewer 2 Report
The authors improved the manuscript according to my comments and suggestions. All of the figures are easily viewable. And also, the revised manuscript in English is better. I am looking forword to your next manuscript concerning signaling pathways.